# Major sea level fall during the Pliocene M2 glaciation

Zifei Yang [1] ✉, Caroline H. Lear [1], Stephen Barker[1], Jonathan Elsey [1], Edward Gasson[2], Yair Rosenthal[3], Sophie M. Slater [1] & Amy Thomas-Sparkes[1]

The extent of ice growth during the Pliocene M2 glaciation (~3.3 Ma) has been called into question, with benthic foraminiferal oxygen isotope records interpreted primarily as a cooling signal. Here we improve the benthic foraminiferal Mg/Ca paleothermometer, allowing bottom water temperature reconstructions with a precision of ±0.2-0.3°C (1 s.d.). Applying this approach to M2 implies a significant increase in ice volume (~55 m SLE) that was more tightly coupled to a drop in $CO_2$ than to ocean temperature. We suggest that the M2 glaciation was driven by a reduction in northern hemisphere poleward heat transport, and amplified by a reduction in southern hemisphere poleward heat transport caused by restriction of the Indonesian Seaway. The cryosphere growth drove the atmospheric $CO_2$ decrease, which likely contributed to the overall magnitude of ice growth. These results demonstrate the sensitivity of the cryosphere to changes in ocean heat transport in a similar to modern climate.

Geological records of past warm periods in Earth's history provide a window to the behaviour of the Earth system, including the sensitivity of ice sheets to elevated $CO_2$. The warm climate of the mid-Pliocene was interrupted by a short-lasting (~25 kyr) but intense cooling event, Marine Isotope Stage (MIS) M2, which has been thought of as an early but failed initiation of Northern Hemisphere glaciation, based on records of benthic foraminiferal $\delta^{18}O$, $\delta^{13}C$, and ice-rafted debris (IRD)[1–4]. There is no consensus on the magnitude of cooling versus ice volume growth across the M2 glaciation[5,6], nor the trigger or forcing factors involved[7–9]. Due to high levels of atmospheric $CO_2$ (~300–500 ppm), it is thought that insolation forcing alone would not have been able to drive major ice sheet formation at M2[10,11]. De Schepper et al.[7] suggested that the opening and closing of the shallow Central American Seaway (CAS) might have played an important role in the onset and end of the M2 glaciation. But modelling experiments suggest that opening of the CAS could lead to at most a 16 m sea-level fall, which corresponds to the minimum estimate from proxy reconstructions[8]. Further inferences about the M2 glaciation event are currently impeded by the high level of uncertainty and low temporal resolution of published proxy records[5,6]. Past changes in glacio-eustatic sea level

have been particularly challenging to reconstruct with useful precision[12]. Benthic foraminiferal $\delta^{18}O$ contains the signals of both ice-volume and bottom water temperature (BWT). Therefore, to reconstruct ice-volume change using this approach, BWT must be quantified. Benthic foraminiferal calcite Mg/Ca has been widely used in calculating BWT[13,14]. However, when investigating the relatively small glacial cycles of the early and mid-Pliocene, the uncertainty of the BWT reconstruction can lead to uncertainties in sea-level reconstruction larger than the glacial-interglacial signal itself[12,15–17]. The uncertainties in reconstructed BWT stem from factors such as changing seawater Mg/Ca, carbonate saturation state ($\Delta CO_3^{2-}$), and importantly, the scatter in modern core-top calibrations used to define the sensitivity of foraminiferal Mg/Ca to BWT[18].

The infaunal foraminifer genus *Melonis* spp. represents a potential solution for current issues in Mg/Ca-BWT reconstruction. Firstly, infaunal foraminifera are expected to be less affected by changes in bottom water $\Delta CO_3^{2-}$ than epifaunal species[15,16]. Secondly, the temperature calibration for *Melonis* spp. has unusually good precision[19]. But there are still some poorly constrained factors in the application of *Melonis* Mg/Ca-palaeothermometry, including the sensitivity of

[1]School of Earth and Environmental Sciences, Cardiff University, Cardiff, UK. [2]Department of Earth and Environmental Sciences, University of Exeter, Penryn, UK. [3]Department of Marine and Coastal Sciences and Department of Earth and Planetary Sciences, Rutgers the State University, New Brunswick, USA. ✉ e-mail: Zifei.Yang@soton.ac.uk

*Melonis* Mg/Ca to porewater $\Delta CO_3^{2-}$, and how test size (ontogenetic effect) might affect Mg/Ca and $\delta^{18}O$ values. To address these issues, we measured multiple proxies ($\delta^{18}O$, $\delta^{13}C$, Mg/Ca, B/Ca) in *Melonis* specimens picked from two size fractions (150–250 μm and 250–355 μm) in down-core samples from ODP Sites 982 and 1241, and core-top samples from the Norwegian Sea, Little Bahama Bank and the Indonesia Seaway. Our results yield an approximately three-fold improvement in the uncertainty of sea level reconstructions using the Mg/Ca-$\delta^{18}O$ palaeothermometry approach, which transforms its utility in investigating cryospheric processes. We apply this approach to generate high-resolution (-2 kyr) records of bottom water temperatures and global ice volume from 3.313 Ma to 3.184 Ma (MIS M2) and a comparison interval from 4.421 Ma to 4.337 Ma. Our results overturn recent interpretations of the M2 event as a predominantly cooling event. Instead, our records from two sites both reveal a significant ice volume signal, which is consistent with ice growth in both hemispheres. We discuss the possible causes for the onset and termination of the M2 glaciation against a backdrop of similar to modern $CO_2$ concentrations.

## Results and discussion
### Improving the benthic Mg/Ca palaeothermometer
Infaunal benthic foraminiferal species such as *Uvigerina* spp., *O. umbonatus*, and *Melonis* spp. are typically preferred for Mg/Ca paleothermometry as they are thought to be less affected by changes in bottom water $\Delta CO_3^{2-}$ compared with epifaunal species such as *Cibicidoides* spp.[15,19,20]. However, core-top *Melonis* spp. from regions with high sedimentary total organic carbon (TOC) have unusually low Mg/Ca, which suggests that porewater chemistry may potentially affect *Melonis* Mg/Ca[19]. Here, we use the $\delta^{18}O$, $\delta^{13}C$, and element/Ca ratios in *Melonis* from different test size fractions to discuss the effects of microhabitat on foraminiferal calcite test geochemistry. The chemical composition of an infaunal foraminifer test depends on the chemical composition of porewater, and influences such as temperature, $\Delta CO_3^{2-}$, and calcification rate[21]. *Melonis* is an intermediate infaunal species that usually lives within 8 cm of the seafloor, with variable average habitat depths of 1–4 cm at different locations[22–24]. Over these small depth scales, we can ignore temperature changes within its depth habitat[25], and focus on the influence of chemical or ecological gradients within the microhabitat, and ontogenetic effects on *Melonis*.

The $\delta^{18}O$ values of different size fractions among the study sites show no systematic differences (Fig. 1a), with the linear regression equation of $\delta^{18}O_{M\ (250-355\ μm)} = 0.98 \times \delta^{18}O_{M\ (150-250\ μm)}$ ($r^2 = 0.999$, $n = 24$). At ODP Site 982, the $\delta^{13}C$ values of 150–250 μm *Melonis* spp. are generally higher than the 250–350 μm *Melonis* spp. (Fig. 1c). At ODP Site 1241 and the Norwegian Sea stations, more than half of the samples are located around the 1:1 line, while the rest of the samples indicate higher values of $\delta^{13}C$ of 150–250 μm *Melonis* spp relative to the 250–350 μm *Melonis* spp. (Fig. 1c). Ontogenetic increases of $\delta^{18}O$ and $\delta^{13}C$ with test size have been reported for some buliminid taxa, including species of genera *Uvigerina*, *Bolivina*, *Bulimina*, and *Globobulimina*, while rotaliid taxa, including species of genera *Melonis* and *Cibicidoides,* have been shown to have negligible size-related effects[26,27]. This lack of an ontogenetic effect on the stable isotope composition of *Melonis* calcite is consistent with our results that show no offset in $\delta^{18}O$ between the two size fractions across a range of sites (Fig. 1a). By extension, we consider it most likely that the offset in $\delta^{13}C$ between the two size fractions at some sites (Fig. 1c) reflects differences in porewater chemistry rather than ontogeny. Studies of the influence of microhabitats on benthic foraminiferal $\delta^{13}C$ report that epifaunal species record the bottom water dissolved inorganic carbon (DIC) $\delta^{13}C$, while infaunal species reflect the $\delta^{13}C$ of surrounding porewaters[28–31]. Within the upper 10 cm of the oceans (Pacific and Atlantic) sediments with a water-depth range of 795–4910 m and TOC content of 0.23–6%, decomposition of organic matter in the sediments

leads to a continuous decrease of DIC $\delta^{13}C$ with sediment depth (Fig. 2a)[32–36]. The $\delta^{13}C$ of infaunal foraminifera reflects the porewater $\delta^{13}C$ at their average calcification depth, with lower $\delta^{13}C$ therefore indicating a deeper calcification depth[27,29,31]. The sediment depth habitat of *Melonis* can be variable and is potentially affected by food supply[24]. In this study therefore, the most likely explanation for the samples that have higher $\delta^{13}C$ values in the small size fractions is, at those certain time and geographical points, *Melonis* spp. with test size of 150–250 μm had a shallower average sediment depth habitat than those from the 250–355 μm size fraction. In support of our interpretation, studies of live infaunal species *Melonis* spp. and *Uvigerina* spp. in the modern northern Arabian Sea[37], Sulu Sea[38] and Mediterranean Sea[27,29] also found that individuals with larger test sizes had a deeper average habitat depth.

Porewater $\Delta CO_3^{2-}$ reflects the $\Delta CO_3^{2-}$ of overlying bottom water, which is modified in the sediment column primarily by the decomposition of organic matter. Although organic matter decomposition rates vary spatially and temporally, the average values of porewater $\Delta CO_3^{2-}$ ($\geq 1 \leq 10$ cm) strongly correspond to bottom water $\Delta CO_3^{2-}$ (Fig. S1)[39]. Porewater $\Delta CO_3^{2-}$ profiles consistently display a trend to lower levels of saturation within the first -1 cm of the sediment[39]. However, trends below -1.5 cm sediment depth vary between sites. In general, at Atlantic sites with water depths of 605–4000 m, porewater $\Delta CO_3^{2-}$ is either stable or increases up to 10 cm sediment depth. Conversely, at sites with water depths of 4000–5000 m, porewater $\Delta CO_3^{2-}$ is either stable or decreases up to 10 cm sediment depth (Fig. 2)[39]. Since average porewater $\Delta CO_3^{2-}$ values have a linear relationship with the $\Delta CO_3^{2-}$ of bottom waters (Fig. S1)[39], bottom water $\Delta CO_3^{2-}$ has been used to estimate the effects of porewater $\Delta CO_3^{2-}$ on the geochemistry (e.g., trace metal composition) of infaunal foraminifera. For example, a global core-top study found that B/Ca ratios of strictly epifaunal species (*C. wuellerstorfi*), epifaunal species (*C. mundulus*, *C. robertsonianus*, *Planulina ariminensis*), infaunal species (*Melonis* spp., *Uvigerina* spp., *Oridorsalis umbonatus*, *Gyrodina soldanii*, *Ammonia beccarii*, *Lenticulina vortex*), and aragonitic species (*Heoglundina elegans*) are linearly correlated with bottom water $\Delta CO_3^{2-}$ (Fig. S2)[40,41]. This previous work therefore demonstrated that, in common with many other benthic species, *Melonis* spp. B/Ca increases with increasing calcite saturation state (albeit with a relatively low sensitivity) (Fig. S2). Some of our *Melonis* spp. samples from ODP Sites 982 and 1241 show higher B/Ca ratios in the 250–355 μm size fraction than the 150–250 μm size fraction, with an average offset of $8.3 \pm 10.6$ μmol/mol (Fig. 1d). If, as argued above, we assume that the individuals from the larger size fraction tend to live at deeper average depths, this could imply that either porewater $\Delta CO_3^{2-}$ increased between 1 and 10 cm sediment depth at these sites, and/or that there is an independent ontogenetic or growth rate effect that results in a positive relationship between B/Ca and test size. Sites 982 and 1241 are both shallower than 2500 m (Table S1 and Fig. S3), so comparison with modern porewater profiles suggests that porewater $\Delta CO_3^{2-}$ was likely either stable or increasing between 1 and 10 cm depth in the sediment (Fig. 2b, c). Therefore, while we do not rule out secondary biological effects, we prefer the simplest explanation, which is that in our samples, larger individuals of *Melonis* spp. tended to live slightly deeper in the sediment, experiencing on average lower $\delta^{13}C$ and higher $\Delta CO_3^{2-}$. The average habitat depth and associated porewater $\Delta CO_3^{2-}$ experienced by *Melonis* spp. will vary spatially and temporally. However, it is important to note that the combination of porewater chemistry and possible secondary biological effects have not impacted *Melonis* Mg/Ca. Our evidence for this assertion is that there is no systematic difference in measured Mg/Ca between the two size fractions, with a linear regression equation of Mg/Ca$_{(250-355\ μm)} = 0.999 \times$ Mg/Ca$_{(150-250\ μm)}$ ($r^2 = 0.999$, $n = 34$) (Fig. 1). This observation suggests that *Melonis* spp. has a calcification mechanism that enables it to calcify in variable $\Delta CO_3^{2-}$/Ca. We note that Hasenfratz et al.[19] attributed

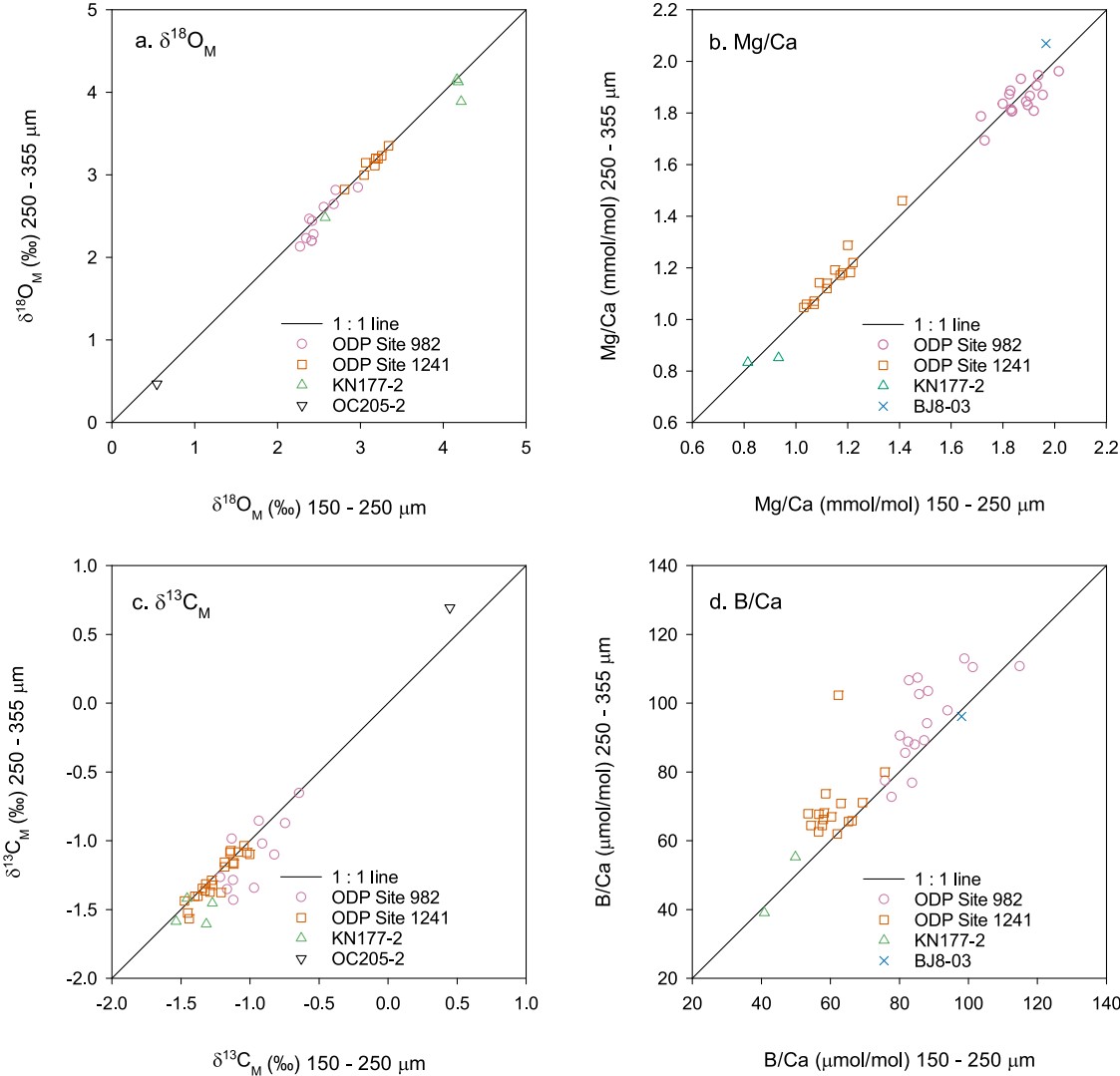

**Fig. 1 | Stable isotope values and trace metal compositions in *Melonis* spp. from different size fractions. a** $\delta^{18}O$ ($\delta^{18}O_M$), **b** Mg/Ca, **c** $\delta^{13}C$ ($\delta^{13}C_M$), **d** B/Ca, in the 150–250 μm size fraction versus the 250–355 μm size fraction of *Melonis* spp. from the same samples. Solid lines–1:1 line; Pink circles–ODP Site 982 down-core samples; Orange squares–ODP Site 1241 down-core samples; Green up triangles–Site KN177-2 near core-top samples; Black down triangles–Site OC205-2 core-top sample; Blue crosses–BJ8-03 core-top sample.

anomalously low *Melonis* spp. Mg/Ca from tropical Atlantic sediments with high TOC to either a carbonate saturation state effect on the incorporation of Mg into the calcite test, or a post-mortem dissolution effect. Hasenfratz et al.[19] considered the first option more likely, since the specimens showed no apparent dissolution features. However, if this was the case, we would expect to see a relationship between Mg/Ca or Mg/Ca residuals and $\Delta CO_3^{2-}$ in Figs. 3b, 3c, which is not the case. Post-mortem preferential dissolution of high Mg/Ca portions of test calcite can lower foraminiferal Mg/Ca without resulting in obvious dissolution under optical microscopes, and we suggest that this is the mechanism for the anomalously low Mg/Ca at the high TOC sites[42–44]. We conclude overall that since *Melonis* spp. $\delta^{18}O$ and Mg/Ca appear to be insensitive to ontogenetic effects and microhabitat $\Delta CO_3^{2-}$, paired $\delta^{18}O$ and Mg/Ca palaeothermometry using *Melonis* spp. is a reliable tool in reconstructing past bottom water temperature and global ice volume, which enables us to refine the calibration and approach for this species. We have added new core-top data to the calibration dataset, removed data likely affected by post-mortem dissolution (high TOC sites; noting therefore that this revised calibration should

be used with caution at high TOC sites), and accounted for our approach which analyses more individuals per sample to reduce the effect of inter-shell variability (Fig. 3; "Methods"). Together, these result in improved uncertainties in the BWT and $\delta^{18}O_{SW}$ estimates, which are ±0.2–0.3°C and ±0.06–0.08 ‰, respectively (1 s.d.; "Methods"). These lower uncertainties are supported by the low variability in our BWT record in the older interval ("Methods"; Fig. 4). In addition, we tested our approach on core top samples from the Norwegian Sea. Calculated $\delta^{18}O_{SW}$ generated (0.39 ± 0.18‰; 1 s.d.) is in very good agreement with measured hydrographic $\delta^{18}O_{SW}$ (-0.32‰[45]). We note that our approach of analysing all samples against matrix-matched standards without diluting samples contributes to our high precision downcore records[20,46].

## Significant ice growth at MIS M2

MIS M2 has been considered the most intense Pliocene glaciation prior to the intensification of northern hemisphere glaciation around 2.7 Ma. However, a recent study using Mg/Ca-palaeothermometry and clumped isotopes on Atlantic IODP Site U1308 (49.87 °N, 24.23 °W;

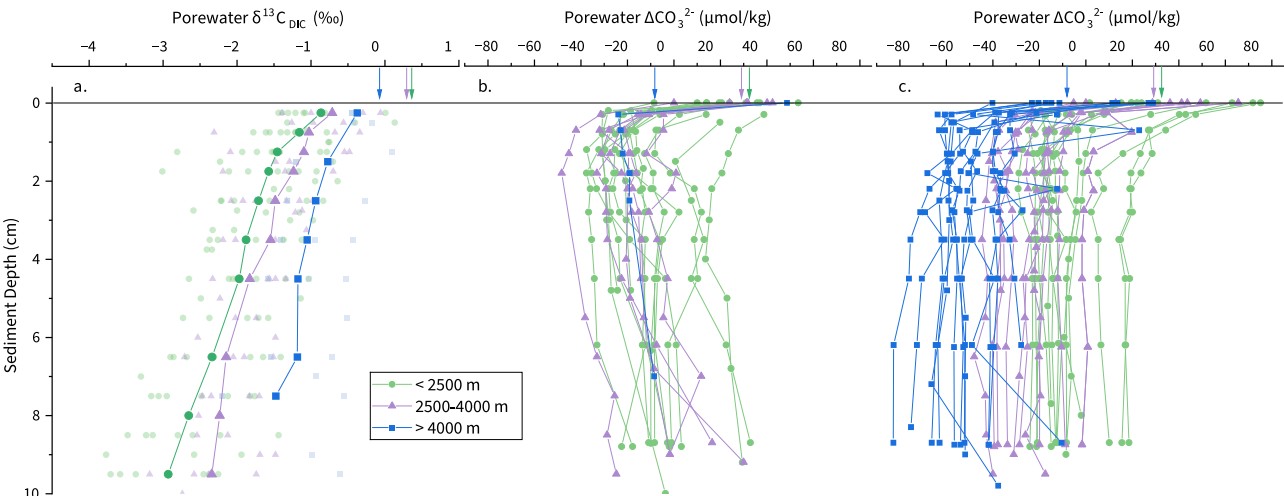

**Fig. 2 | Porewater $\delta^{13}C_{DIC}$ and $\Delta CO_3^{2-}$ profiles through the top 10 cm of the sediment column at globally distributed ocean sites.** Colour coded by water depth of site, water depths <2500 m (green circles), 2500–4000 m (violet triangles) and > 4000 m (blue squares). **a** Porewater $\delta^{13}C_{DIC}$[32–35], pale symbols indicate the measured values at individual stations, bold symbols with lines indicate the average trends of each water-depth group. **b** Porewater $\Delta CO_3^{2-}$ profiles that display a trend of increasing $\Delta CO_3^{2-}$ below ~1.5 cm sediment depth. **c** Porewater $\Delta CO_3^{2-}$ profiles that display a trend of stable values or decreasing $\Delta CO_3^{2-}$ below ~1.5 cm sediment depth. Arrows represent average values of $\delta^{13}C_{DIC}$ and $\Delta CO_3^{2-}$ of overlying bottom water of 3 water-depth groups, respectively.

3871 m) and Pacific ODP Site 849 (0.18 °N, 110.52 °W; 3851 m) has suggested that the global $\delta^{18}O$ signal predominantly reflects bottom water cooling (of ~4 °C) with minimal change in ice volume (see the locations of sites in Fig. S3)[5]. That temperature record was calculated using Mg/Ca of *O. umbonatus*, a shallow infaunal species, which has been shown to be less sensitive to changes in $\Delta CO_3^{2-}$ than epifaunal species[47], but can still be affected by changes in porewater $\Delta CO_3^{2-}$ where sites are not fully buffered[17]. For this reason, it has been recommended that *O. umbonatus* Mg/Ca palaeothermometry is applied on sites with abundant and well-preserved planktonic foraminifera, which reflects a high buffering capacity of the sediment[17]. IODP Site U1308 and ODP Site 849 both sit at depths >3.8 km, and ODP Site 849 has poorly preserved planktonic foraminifera throughout the section. Without accompanying B/Ca records, it is difficult to determine whether *O. umbonatus* at this site experienced more corrosive porewaters during the M2 event, which would likely have lowered Mg/Ca, potentially leading to an overestimation of the cooling. We also note that at both sites, *O. umbonatus* Mn/Ca values are an order of magnitude higher than desired (Supplementary Information and Fig. S4). We therefore consider the contribution of cooling and ice growth to the M2 event to remain an open question.

To reconstruct the magnitude and timing of M2 cooling and ice growth, we analysed *Melonis* spp. $\delta^{18}O$, Mg/Ca, $\delta^{13}C$, and B/Ca from two mid-Pliocene intervals (3.313–3.184 Ma, 4.421–4.337 Ma, Fig. 4). The older interval was analysed at ODP Site 982, and the younger interval includes M2 and was analysed at both ODP Sites 982 and 1241. Individual site records reflect local water mass and global climatic changes, so we chose these sites, bathed by different water masses at different water depths, to improve our interpretation of global climatic change ("Methods"; Fig. 4).

The M2 glaciation is represented by a positive and steep $\delta^{18}O_M$ excursion of ~0.66‰ and ~0.59‰ at ODP Sites 982 and 1241, respectively (Fig. 4a). During this period, Mg/Ca (i.e. BWT) decreased at both sites to a minimum at 3.297 Ma, then increased towards the end of MIS M2 (Fig. 4b). The Mg/Ca amplitudes are ~0.28 and ~0.24 mmol/mol, and applying our improved *Melonis* spp. Mg/Ca paleothermometer ("Methods") reveals a 2.6 °C and 2.2 °C cooling at ODP Sites 982 and 1241, respectively. We calculated $\delta^{18}O_{SW}$ using paired $\delta^{18}O_M$ and *Melonis* Mg/Ca-BWT ("Methods"; Fig. 4c). $\delta^{18}O_{SW}$ increased at the start of M2, reaching a maximum at the end of M2 (~3.287 Ma) with an amplitude of ~0.6‰ at both sites. After MIS M2, $\delta^{18}O_{SW}$ decreased to pre-MIS

M2 values. Consistent with our smaller calculated uncertainties on both BWT and $\delta^{18}O_{SW}$, our records are significantly less noisy than the published *O. umbonatus* Mg/Ca records, which we attribute to a combination of factors, including the increased number of individuals analysed and smaller secondary effects such as impacts of variable $\Delta CO_3^{2-}$ and authigenic coatings (Fig. 5). Of note is the observation that during the M2 event, cooling and ice growth (as recorded by $\delta^{18}O_{SW}$) were not coincident and therefore appear decoupled: Changes in BWT led changes in $\delta^{18}O_{SW}$ by about 10 kyr at both sites, which resulted in a ~3 kyr lag of $\delta^{18}O_M$ relative to BWT (Fig. 6). The lag of $\delta^{18}O_{SW}$ to BTW reflects the slow response of ice sheet to forcings, and the $\delta^{18}O_M$ reflects the combined influence of slow ice sheet response and faster temperature signal. A similar lead/lag relationship of ice volume and temperature (bottom water, sea surface and atmospheric temperatures) has also been reported from the Holocene to Pliocene[48–51].

Our *Melonis* spp. BWT records from the Atlantic and Pacific sites are similar to an inferred deep-sea temperature record derived from benthic $\delta^{18}O$ and an independent sea level estimate calculated using a planktonic $\delta^{18}O$ record from the Mediterranean Sea (Fig. 5)[52]. Our ODP Site 982 $\delta^{18}O_{SW}$ record also closely matches the independent Mediterranean sea level record in the older interval, which is centred on interglacial MIS CN5 (Fig. 5)[52]. However, there are some subtle differences between our M2 $\delta^{18}O_{SW}$ records and the Mediterranean sea level and benthic foraminiferal $\delta^{18}O$ deconvolved records (Fig. 5)[52,53]. During M2, $\delta^{18}O_{SW}$ at ODP Sites 982 and 1241 increased in two distinct steps, each ~0.3‰, centred at 3.293 Ma and 3.287 Ma (Figs. 5 and 6). The Rohling et al.[52] $\delta^{18}O_{SW}$ record displays only one increase of ~0.3‰ through M2, which is similar in timing and magnitude to our first step, and which immediately follows the cooling in our BWT record (Figs. 5 and 6). We therefore consider it highly likely that the M2 glaciation was associated with a $\delta^{18}O_{SW}$ increase of at least 0.3‰, which is equivalent to >27 m sea level fall using the $\delta^{18}O_{SW}$-sea level relationship of Fairbanks and Matthews (1978)[54].

However, our second $\delta^{18}O_{SW}$ step is not recorded in the Rohling et al.[52] $\delta^{18}O_{SW}$ record, and furthermore was associated with a warming of bottom waters at both sites. There are two possible reasons for this: firstly, that our *Melonis* spp. $\delta^{18}O_{SW}$ record includes a local salinity signal rather than a global ice volume signal, and secondly, the temporal resolution of the Rohling et al.[52] $\delta^{18}O_{SW}$ record is not sufficient to capture the second rapid and transient (7-10 kyr) increase in ice volume. For example, it is possible that the first sea-level fall of the M2

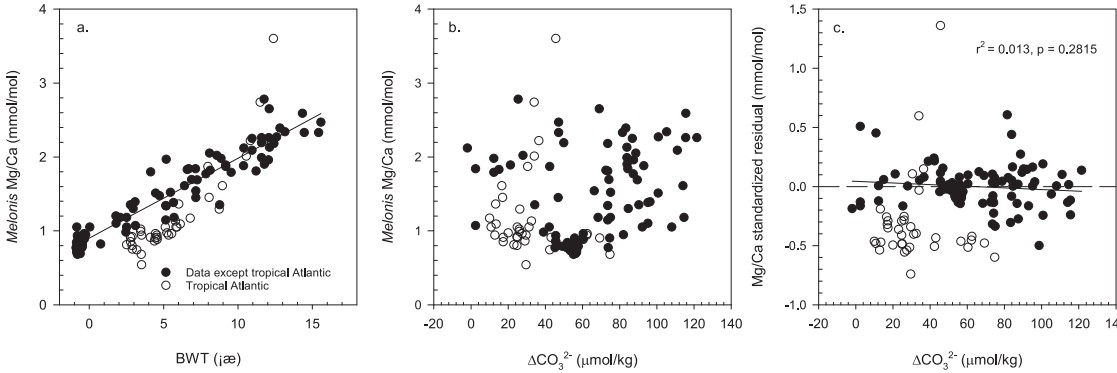

**Fig. 3 | *Melonis* Mg/Ca compared to bottom water temperature and $\Delta CO_3^{2-}$.**
**a** Revised linear calibration between *Melonis* spp. Mg/Ca and bottom water temperature (BWT). **b** Mg/Ca versus bottom water $\Delta CO_3^{2-}$. **c** Mg/Ca standardised residual versus bottom water $\Delta CO_3^{2-}$. Core-top samples from the Nordic Seas, North Atlantic, Oslofjord, Eastern South Atlantic, Sulawesi margins, Pacific Southern Ocean are represented by solid circles, and tropical Atlantic samples are represented by open circles. Including published data and data from this study[15,18,19,81–84]. Solid lines are regression lines.

glaciation restricted water exchange through the Central American Seaway (CAS), subsequently introducing warm and saline water masses to the high latitude Northern Hemisphere[2]. In this scenario, the increase in $\delta^{18}O_{SW}$ would reflect a regional increase in salinity of the bottom water mass rather than a global ice volume signal. However, it is difficult to reconcile this interpretation with the very similar temperature and $\delta^{18}O_{SW}$ history recorded in East Equatorial ODP Site 1241, which was bathed by a mixture of Northern Component Water and Southern Component Water[55,56]. The decreasing $\delta^{13}C_M$ and B/Ca during M2 at both sites are considered to reflect porewater chemistry rather than variation of bottom water composition (Figs. 4, S5, S6 and Supplementary Information). Overall, we consider it unlikely that the same magnitude and timing of the second $\delta^{18}O_{SW}$ increase at ODP Sites 982 and 1241 (Fig. 5) was caused by changing salinity of bottom water masses, and instead we interpret this as a second phase of ice growth during the M2 glaciation.

We therefore suggest that the second rapid and short-lived (7–10 kyr) glaciation phase was not captured in the Rohling et al.,[52] Mediterranean sea level record due to the sampling resolution of the underlying planktonic foraminiferal $\delta^{18}O$ record (Fig. S7). This interpretation could be tested by increasing the sampling resolution of that record. Overall, our records reveal a total $\delta^{18}O_{SW}$ increase of 0.6‰ across the M2 glaciation, which we interpret as a global ice volume signal. Using the Pleistocene $\delta^{18}O_{SW}$-sea level relationship[54], this is equivalent to a total sea level fall of around 55 m. A sea level fall of this magnitude would likely involve some component of northern hemisphere glaciation, although the distribution of IRD in the northern hemisphere (primarily sourced from Greenland and Iceland) suggests that the majority of this ice growth was based in Antarctica[3,4,57]. In addition to the IRD evidence, records of aeolian dust flux in the North Atlantic[58], microbial biomarkers at Mojave Desert[59] and vegetation variability at Yermak Plateau, Arctic Ocean[60] also indicate a cold and dry northern hemisphere climate, typical of glacial conditions, during MIS M2. The sea level fall occurred in two phases, and the high resolution and precision of our records enable us to explore potential mechanisms for the ice growth and retreat in the relatively warm climate of the mid-Pliocene.

## Ocean gateways controlled the M2 glaciation

Our high-resolution records reveal a dynamic cryosphere during the M2 glaciation, with an intriguing relationship with ocean temperature and $p CO_2$. Before comparing our ice volume records with other paleoclimate records in detail, the robustness of the age control was confirmed by the agreement of benthic foraminiferal $\delta^{18}O$ among Sites (Fig. S8). Immediately prior to the first phase of ice growth, a cooling of North Atlantic sea surface temperatures recorded by the alkenone $U^{K'}_{37}$ proxy[61] was also associated with a cooling of Northern Component Water in our ODP Site 982 BWT record (Fig. 6). This cooling of high latitude northern hemisphere sea surface is not seen in sea surface temperature records at ODP Site 1241[62] or Caribbean Site ODP 999[7], and occurred during an interval of rising $CO_2$[10] (Fig. 6). It therefore most likely reflects a regional change in ocean heat transport, perhaps due to the re-opening of the shallow CAS, which weakened the North Atlantic Current (NAC), and reduced the transfer of heat to northern high latitudes[7,8,63]. The gradual cooling of the North Atlantic is mirrored by a gradual shoaling of the lower boundary of Northern Component Water as recorded by the Nd isotope composition ($\varepsilon_{Nd}$) of fish debris at South Atlantic ODP Site 1267 (29 °S, 2 °W; 4350 m)[64], supporting a change in Atlantic Meridional Overturning Circulation through the M2 event (Fig. 6). The high resolution $\varepsilon_{Nd}$ record from North Atlantic IODP Site U1313 (411 °N, 32.4 °W; 3426 m) is less straightforward to interpret, as it is surprisingly decoupled from the South Atlantic ODP Site 1267 $\varepsilon_{Nd}$ record (Fig. 6; see locations of sites and modern Atlantic water masses distribution in Fig. S3)[64]. However, the older changes in the Site U1313 $\varepsilon_{Nd}$ record predate the first sea level fall, and may reflect changes in the source regions of Northern Component Water (Fig. 6 and Fig. S3)[64]. This early northern hemisphere cooling was immediately followed by the first glaciation phase of the M2 event as recorded by our $\delta^{18}O_{SW}$ records, which occurred during an interval of declining northern hemisphere summer insolation (Fig. 6). It therefore seems likely that the tectonic reopening of the CAS preconditioned the system for glaciation, and the precise timing was set by a favourable orbital configuration, which led to a series of cool summers in the northern hemisphere.

It has previously been suggested that the M2 sea level fall led to further restriction of the Indonesian Throughflow (ITF), reducing meridional heat transport to high southern latitudes, and hence amplifying the M2 glaciation by promoting growth of the Antarctic Ice Sheet[65,66]. We suggest that this amplification mechanism was initially relatively small due to unfavourable atmospheric $CO_2$ and orbital configuration (Fig. 6). However, the second sea level fall occurred during a southern hemisphere summer insolation minimum, and by this time $CO_2$ had also fallen to ~350 ppm (Fig. 6). We propose therefore that these three factors (ITF restriction, insolation minima, reduced radiative forcing) combined to provide the "perfect storm", facilitating the particularly intense M2 glaciation (total sea level fall ~55 m SLE). Interestingly, the second sea level fall of the M2 glaciation appears to coincide with a short-lived incursion of Southern Component Water at deepwater North Atlantic ODP Site U1313 (Fig. 6)[64]. This observation is consistent with the growth of the Antarctic Ice Sheet leading to enhanced formation of cold, dense bottom waters in the Southern Ocean.

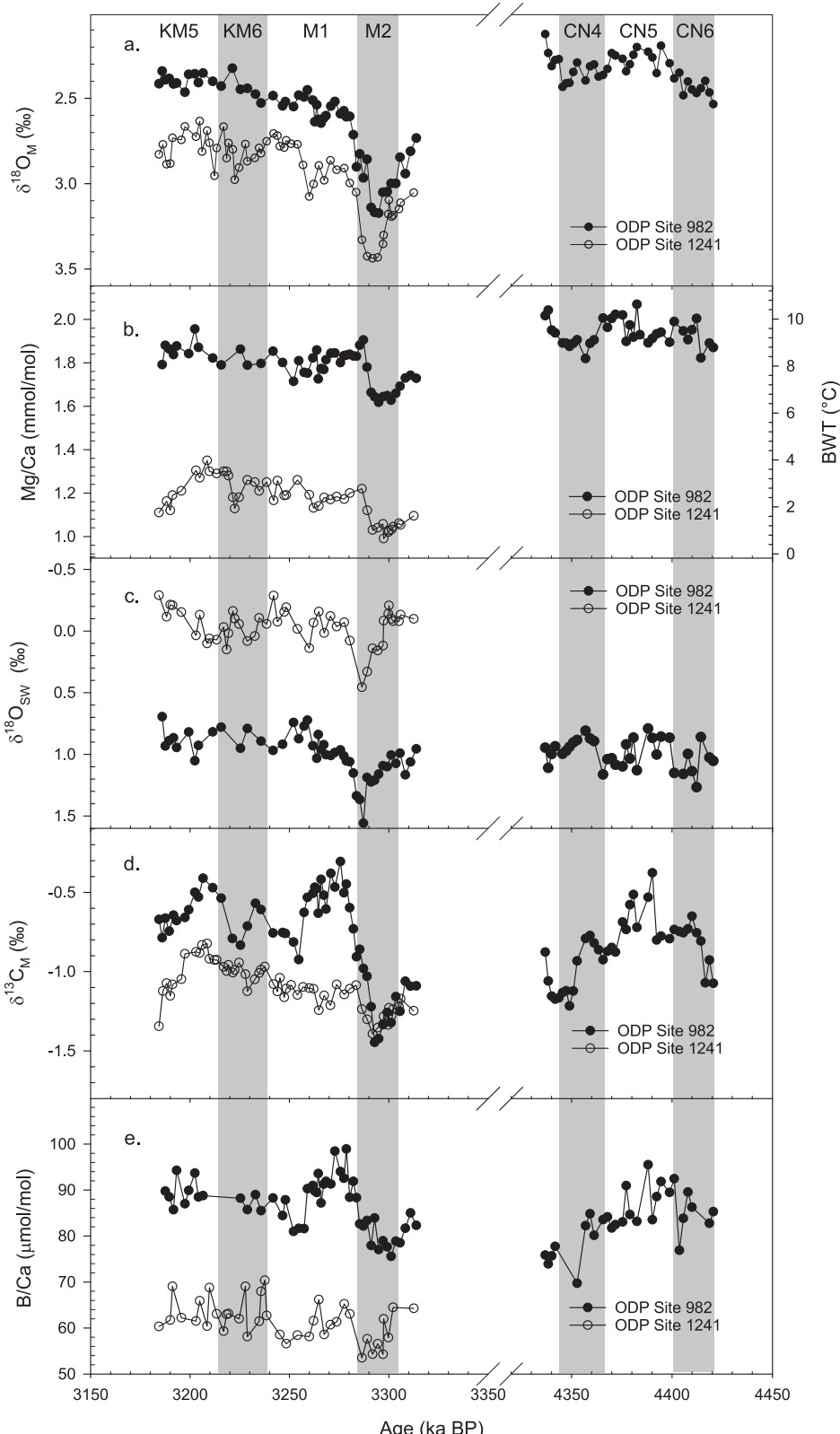

**Fig. 4 | Down core geochemical records at ODP Sites 982 and 1241.** Solid circles represent ODP Site 982 records, open circles represent ODP Site 1241. **a** *Melonis* $\delta^{18}O$ ($\delta^{18}O_M$), **b** Mg/Ca and bottom water temperature (BWT), **c** reconstructed seawater $\delta^{18}O$ ($\delta^{18}O_{SW}$), **d** *Melonis* $\delta^{13}C$ ($\delta^{13}C_M$) and **e** *Melonis* B/Ca. Gery bands show the glacial periods, including MIS KM6, M2, CN4 and CN6. Interpretation of $\delta^{13}C_M$ and B/Ca records is provided in the Supplementary Information.

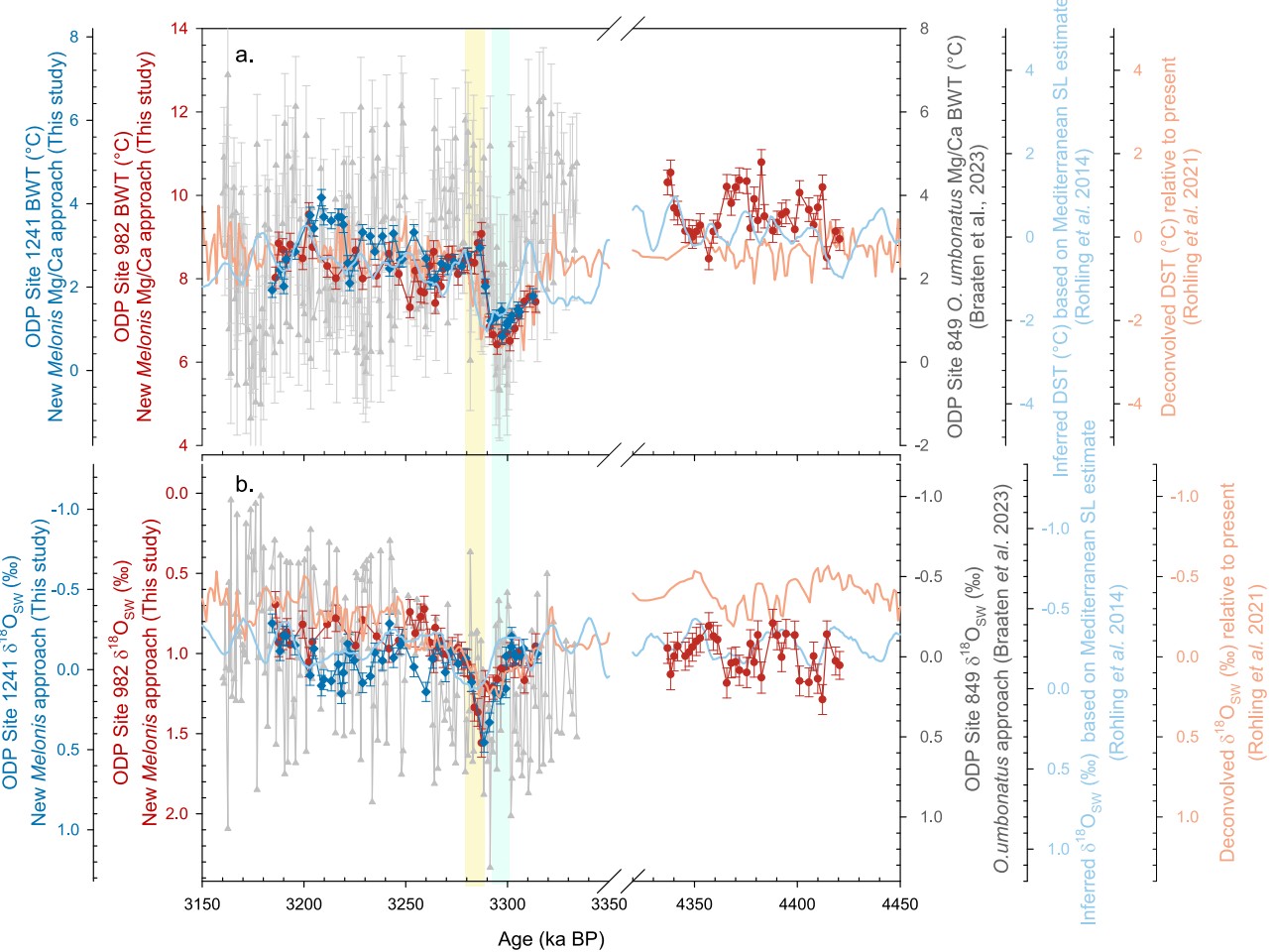

**Fig. 5 | Bottom water temperature and seawater $\delta^{18}O$ records reconstructed using *Melonis* spp. approach compared with published records at Mid-Pliocene. a** bottom water temperature (BWT) or deep sea temperature (DST) and **b** seawater $\delta^{18}O$ ($\delta^{18}O_{SW}$) records at ODP Sites 982 (red circles) and 1241 (blue diamonds) reconstructed using the improved *Melonis* spp. approach in this study, published proxy records using the *O. umbonatus* Mg/Ca approach (Grey triangles)[5], model-based records of inferred records based on the Mediterranean sea level estimate (Light blue lines)[52] and deconvolved records based on the LR04 stack (Orange lines)[53]. The green and yellow bands show the first and the second phases of glaciation at ODP Sites 982 and 1241, respectively. Error bars represent ± 1 s.d.

It is striking that high latitude northern hemisphere temperatures increased following the first sea level fall (Fig. 6). It is possible that the second sea level fall restricted the CAS, reinvigorating the NAC, and transporting heat to the north once more, although the gateway was likely also affected by ongoing tectonic changes[67]. Approximately halfway through this regional warming, global ice volume started to retreat, eventually reaching pre-M2 levels by 3.276 Ma. We postulate that the deglaciation phase of MIS M2 was affected by a similar series of oceanographic feedbacks. For example, the planktonic foraminiferal records from De Vleeschouwer et al.[65] and nannofossil assemblage data from Auer et al.[66] indicate that the end of MIS M2 was associated with an enhanced ITF, which facilitated heat transport to high southern latitudes once more.

It is interesting to note that the timing of changes in atmospheric $CO_2$ is more similar to those in our calculated record of $\delta^{18}O_{SW}$ than the BWT and sea surface temperature (SST) records (Fig. 6)[5]. Our record of $\delta^{18}O_{SW}$ reveals that the M2 ice growth occurred between 3.3 and 3.287 Ma, during which time $CO_2$ decreased from ~466 to ~312 ppm, whilst sea surface temperatures and bottom water temperatures were either stable or increasing (Fig. 6). Therefore, we consider it more likely that the M2 glaciation was primarily caused by changes in ocean heat transport, while the $CO_2$ variations reflect a positive climate

feedback to the cryosphere development. However, high-resolution $CO_2$ records from different ocean basins are required to fully investigate the carbon cycle-climate interactions during the M2 glaciation. This explanation is also supported by previous M2 studies. For example, de la Vega et al.[10] found an apparent lag of $CO_2$ relative to benthic $\delta^{18}O$, and proposed a role for the Southern Ocean carbon cycle in regulating $CO_2$. Hou et al.[68] presented a multi-proxy reconstruction in the subantarctic zone, and found that $CO_2$ variations were more tightly coupled with the position of the subtropical front than $\delta^{18}O$ or sea surface temperatures, further supporting a mechanism linking cryosphere development with the extent of Southern Ocean outgassing of $CO_2$. The M2 glaciation has previously been proposed as a failed attempt at Northern Hemisphere Glaciation, but this study reveals a surprising sequence of events that suggests that its drivers were relatively unique. The M2 glaciation was unusual in that its onset was associated with some northern hemisphere ice growth under moderate $CO_2$ forcing, whilst its subsequent culmination was associated with Antarctic ice sheet growth under reduced $CO_2$ forcing. It highlights the importance of tectonic boundary conditions in understanding glacial transitions, but nevertheless provides a valuable test bed for modelling ice-ocean-atmosphere interactions and feedbacks[69].

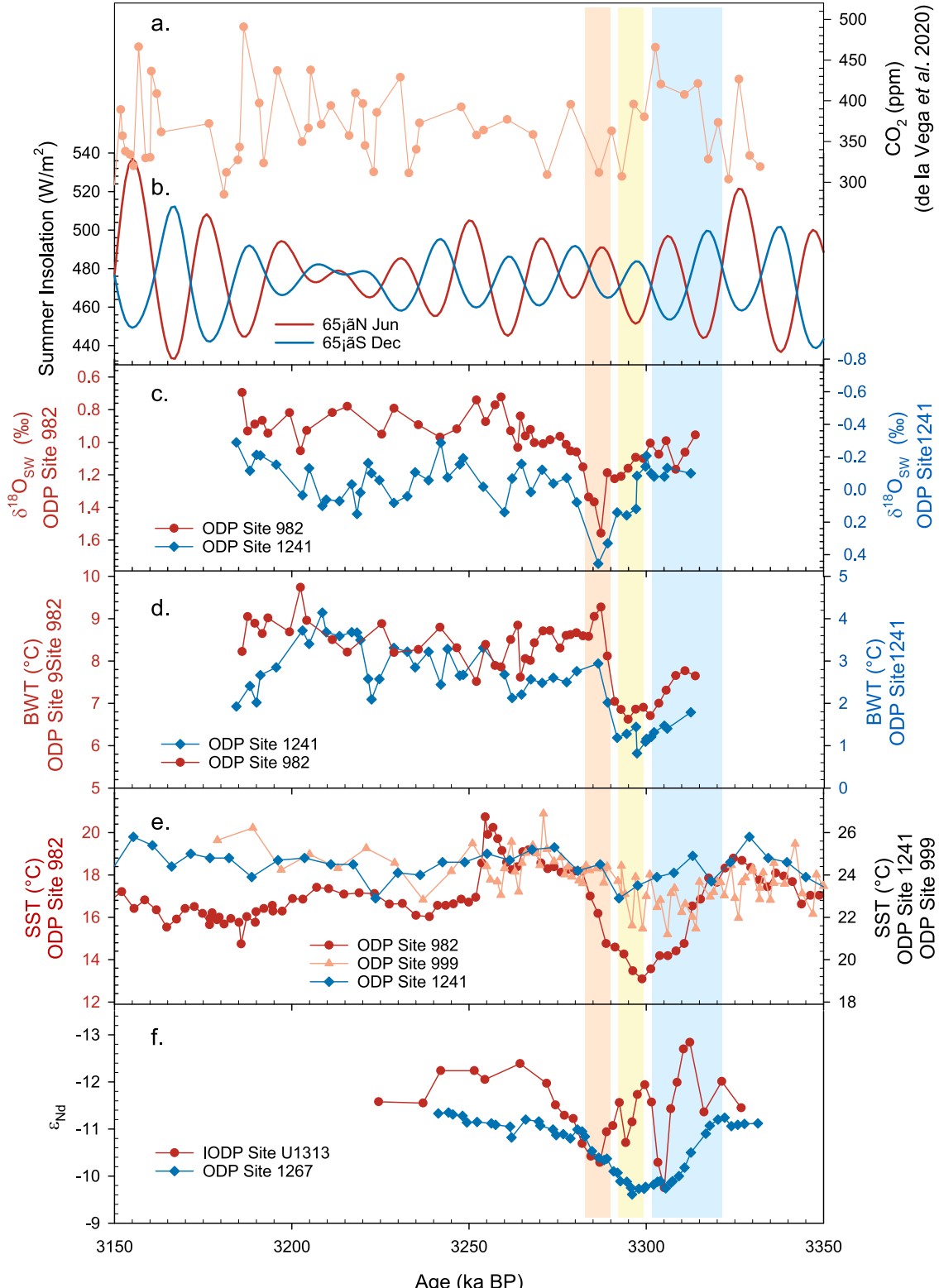

**Fig. 6 | Comparison of geochemical records with insolation and atmospheric CO₂ across the mid-Pliocene M2 glaciation. a** Atmospheric $CO_2$ derived from $\delta^{11}B$ of *G. ruber* at ODP Site 999[10]. **b** Summer insolation at 65 °N (June; red line) and 65 °S (December; blue line)[11]. **c** Seawater $\delta^{18}O$ ($\delta^{18}O_{SW}$) at ODP Sites 982 (red circles) and 1241 (blue diamonds). **d** *Melonis* spp. Mg/Ca bottom water temperatures (BWT) at ODP Sites 982 (red circles) and 1241 (blue diamonds). **e** Sea surface temperature (SST) at ODP Site 982 (red circles)[61], 1241 (blue diamonds)[62] and ODP Site 999 (orange triangles)[7]. **f** Fish debris Nd isotope data ($\varepsilon_{Nd}$) at IODP Site U1313 (red circles) and ODP Site 1267 (blue diamonds)[64]. The blue band highlights the decreasing BWT aligned with the cooling of northern hemisphere SST, and the increasing $\varepsilon_{Nd}$ at ODP Site 1267 while the atmospheric CO₂ was relatively high; The yellow band highlights the first phase of sea level fall aligned with a northern hemisphere summer insolation minima and decreasing atmospheric CO₂; The orange band highlights the second phase of sea level fall aligned with a southern hemisphere summer insolation minima and relatively low atmospheric CO₂.

## Methods

### Study sites and chronology

We analysed *Melonis* spp. from ODP Sites 982 and 1241 (see locations in Figs. S3), and 6 sites for near core-top and core-top modern samples (with sediment depth of 2–4 cm of samples from Norwegian Sea and 0–2 cm of all other core-top samples). The specific information of the sites is shown in Table S1 with the modern bottom water temperature (BWT); modern BWT values of ODP Sites 982 and 1241 are from GLO-DAP datasets[70]. We mixed *M. barleeanum* and *M. pompilioides* for trace metal analysis because these two species and their morphotypes have similar responses of Mg/Ca to temperature[19], and we used *M. barleeanum* for stable isotopes. All sites were used to improve the paired benthic foraminiferal $\delta^{18}$O and Mg/Ca-palaeothermometry method, which was then used to generate a series of mid-Pliocene records from ODP Sites 982 and 1241. The age model for ODP Site 982 was developed by Lisiecki and Raymo (2005)[1], while the age model for ODP Site 1241 is based on Tiedemann et al.[71]. Two intervals were chosen: 3.313–3.184 Ma, which includes MIS KM5, KM6, M1 and M2, and 4.421-4.377 Ma, which includes MIS CN4, CN5, CN6. The sample resolution of these downcore records is around 2 kyr. ODP Site 982 is located at a water depth of 1134 m on the Rockall Plateau, bathed in North Atlantic Intermediate Water, consisting of Labrador Sea Water, Modified East Icelandic Water and Mediterranean Overflow Water (Fig. S3)[72]. ODP Site 1241 is situated at a depth of 2027 m on the north slope of the Cocos Ridge in the eastern equatorial Pacific (Fig. S3), and is currently bathed by mid-depth waters of North Pacific origin[73].

### Analytical methods

Bulk sediment samples were soaked in DI water and spun overnight, washed over 63 µm sieves, then dried. *Melonis* spp. were picked separately from the 150–250 µm and 250–350 µm size fractions. Where possible, 15 tests from the 150–250 µm size fraction were analysed for their stable isotope composition, and 35 tests from the 150–250 µm size fraction were analysed for their trace metal composition. When fewer tests were available in a sample, the 150–250 µm size fraction was prioritised for trace metal analysis, and 5–8 specimens from 250–350 µm were used for stable isotope analysis. For some samples, stable isotopes and trace metals were analysed on both size fractions to evaluate any possible size effect on the measured geochemistry. For these samples, 8–20 specimens from the 250–350 µm size fraction were used for the trace metal analysis.

For the $\delta^{18}$O and $\delta^{13}$C analysis, foraminiferal tests were crushed against glass slides to open the chambers, and the fragments were put into acid-cleaned microcentrifuge tubes. These were ultrasonicated three times in DI water, twice in methanol, and another two times in DI water to remove clays. 20–80 µg of each sample was analysed by Gas Isotope Ratio Mass Spectrometer Thermo MAT 253 with Kiel IV Carbonate Preparation Device at Cardiff University. The results are reported versus Vienna Peedee belemnite via standard BCT63 (Carrara marble), and the standard deviation is 0.033‰ for $\delta^{18}$O, and 0.027‰ for $\delta^{13}$C.

The cleaning procedure for trace metal analysis is adapted from Boyle and Keigwin (1985)[74]. We applied the procedure with and without a reductive step on selected horizons from down-core samples from ODP Sites 982 and 1241. The results of this cleaning test (Supplementary Information Table S2) show that the reductive step introduced a ~47% loss of the calcite test. On average, the Mn/Ca ratio is 25% lower, and the Mg/Ca ratio is 4% lower when the reductive step was applied. The 4% decrease of Mg/Ca likely comes solely from the preferential dissolution of calcite during the reductive step[75]. Mn/Ca of samples cleaned without the reductive step at ODP Site 982 are generally 45–76 µmol/mol, which is acceptable for trace metal analysis (below 100 µmol/mol)[76]. For ODP Site 1241, Mn/Ca in samples cleaned with and without a reductive step are both above this limit[76] (159–337 µmol/mol) and (206–500 µmol/mol), respectively). The

reductive step is not effective enough in removing Mn-enriched phases, which introduce less than 4% bias in Mg/Ca, but increase the risk of preferential dissolution[75], hence we did not perform the reductive step for the rest of our samples, which are presented in the main figures. Cleaned and dissolved samples were analysed by High Resolution ICP-Mass Spectrometer (Thermo Finnigan Element XR) at Cardiff University, using Ca concentration-matched standards to reduce matrix effects[18,20]. The long-term precisions, based on the analysis of consistency standards, for Mg/Ca is 1%, B/Ca is 4%. The Mg/Ca data were screened using limits of 600 µmol/mol for Mn/Ca, 500 µmol/mol for Al/Ca and 7 mmol/mol for Na/Ca, and the B/Ca data were rejected when the measured sample intensity of $^{11}$B was less than 7 times the measured intensity of the nearest blank in the sequence.

### Refining the Mg/Ca-temperature calibration

The calibration of *Melonis* spp. Mg/Ca ratio to BWT is based on Hasenfratz et al.[19], which results in average calculated uncertainties of ±0.5 °C and ±0.4 °C (1 s.d.) at ODP Sites 982 and 1241, respectively. We made three adjustments to this published calibration, which result in reduced uncertainties. Firstly, we added new core-top data from the Norwegian Sea and Indonesian Seaway into the calibration (Fig. 3). Secondly, a factor of 1.04, based on the with and without reductive step results of *Melonis* spp. in this study (rather than 1.10), was used to correct the Mg/Ca ratios of reductively cleaned samples compiled in Hasenfratz et al.[19]. Thirdly, we exclude the tropical Atlantic data from sites with relatively high levels of TOC. These samples are identified in Hasenfratz et al.[19] as having anomalously low Mg/Ca, which we attribute to partial dissolution caused by low $\Delta CO_3^{2-}$ resulting from enhanced remineralisation of organic matter. Our Pliocene samples do not have particularly high levels of TOC[73,77], justifying exclusion of these samples from the revised calibration dataset. Our revised linear calibration equation is *Melonis* spp. Mg/Ca (mmol/mol) = 0.108 ± 0.004*BWT (°C) + 0.902 ± 0.026 ($r^2$ = 0.91, $n$ = 91, 1 s.d.; Fig. 3). This calibration predicts average BWT uncertainties of ±0.4 °C and ±0.3 °C (1 s.d.) at ODP Sites 982 and 1241, respectively. However, the average number of specimens used in Hasenfratz et al. (2017) is 13[19], while this study uses generally more than 30, which results in less noisy downcore records by reducing inter-test variability. Applying a factor of $30^{1/2}/13^{1/2}$ leads to a estimated uncertainty of ±0.3 °C and ±0.2 °C (1 s.d.) for our downcore BWT reconstructions at ODP Sites 982 and 1241, respectively. We ground-truth these values by calculating the standard deviation of our BWT records through intervals of stable BWT. During the intervals 4365.7–4375.4 ka and 4384.0–4398.8 ka at ODP site 982, calculated BWT is relatively constant, and the standard deviations of these two intervals are ±0.22 °C ($n$ = 5; 1 s.d.) and ±0.19 °C ($n$ = 8; 1 s.d.) respectively, supporting the low uncertainties derived from the core top calibrations.

**$\delta^{18}$O$_{SW}$ calculation.** In order to calculate the $\delta^{18}$O of seawater ($\delta^{18}$O$_{SW}$), we first corrected the measured *M. barleeanum* calcite $\delta^{18}$O values to calcite equilibrium values $\delta^{18}$O$_{cp}$ by adding 0.276‰[78]. We then used these values and the BWT calculated from Mg/Ca in the *Cibicidoides* oxygen isotope temperature calibration Eq. (1) to calculate $\delta^{18}$O$_{SW}$[79].

$$(\delta^{18}O_{cp} - \delta^{18}O_{SW} + 0.27) = -0.250 \pm 0.005t + 0.0014 \pm 0.002t^2 + 3.56 \pm 0.02 \tag{1}$$

Combined analytical and calibration uncertainties result in average uncertainties for calculated $\delta^{18}$O$_{SW}$ of ±0.08‰ and ±0.06‰ (1 s.d.) at ODP Sites 982 and 1241, respectively. Assuming a $\delta^{18}$O$_{SW}$-sea level relationship of ~0.11‰/10 m SLE[54], this is equivalent to sea level uncertainties on the order of 5–7 m, which is less than half of the uncertainty typically considered achievable with Mg/Ca-$\delta^{18}$O palaeothermometry[12].

## Data availability

All *Melonis spp*. trace metals and stable isotopes data are available via Figshare at https://doi.org/10.6084/m9.figshare.29497142.v1[80], and source data underlying all the figures in the Main Text and also Supplementary Information are provided with this paper. Source data are provided with this paper.

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

## Acknowledgements

This research used downcore samples provided by the Ocean Drilling Programme. This work was funded by Natural Environment Research Council grant NE/T007397/2 to CHL and EG, and also received funding from the European Union's Horizon 2020 research and innovation programme under grant agreement No. 101184070 (P2F) to CHL and SB. We thank Alexandra Nederbragt, Edward Inglis and Lindsey Owen for laboratory assistance. We thank Paul Wilson for useful discussions. This is Cardiff EARTH CRediT contribution 40.

## Author contributions

Z.Y. interpreted the data and wrote the initial draft with support from C.H.L. and S.B. Z.Y., C.H.L., S.B., E.G., Y.R., S.M.S., and J.E. edited and reviewed the final paper. Z.Y., S.M.S., and A.T-S conducted sample analyses. Z.Y. and J.E. conducted the uncertainty analysis. C.H.L. and E.G. procured the funding, and C.H.L. conceived the idea and led the project.

## Competing interests

The authors declare no competing interests.
