## [Transparent Peer Review file · Nature Communications]

Major sea level fall during the Pliocene M2 glaciation

Corresponding Author: Dr Zifei Yang

Version 0:

Reviewer comments:

Reviewer #1

(Remarks to the Author)

All high resolution (<5 kyr) benthic foraminiferal $\delta^{18}\text{O}$ records including the LR04 and the CENOGRID exhibit prominent positive excursion during Marine Isotope Stage (MIS) M2 but rapidly rebound to interglacial level thereafter. Additionally, abundant Ice Rafted Debris (IRD) in the deep-sea sediments of north Atlantic occurred at about 2.7 Ma, much later than MIS M2. A consensus on the timing of the late Pliocene intensification of the Northern Hemisphere Glaciation (iNHG) was thought to be at ~2.7 Ma. Therefore, MIS M2 was traditionally thought to be a failed initiation of NHG. Some studies based on bottom water temperature reconstruction revealed that M2 was only a cooling event, which was not linked with significant sea level fall or prominent polar ice sheet expansion.

My overall impression on this manuscript after careful reading is that this study challenges the traditional understanding on the M2 as mentioned above. Mainly based on an improved benthic foraminiferal Mg/Ca paleothermometer, this study pointed out that global sea level had experienced a two-stage drop reaching as many as 55 meters in total during MIS M2.

The innovation of this study is the improved benthic Mg/Ca paleothermometer that was used to reconstruct the bottom water temperature changes, with improved uncertainties of 0.2-0.3 °C. The Cardiff group has made important contributions to the development of the benthic foraminiferal paleothermometer, such as a pioneer work published in Science in 2000. The excellent relationship shown in Figure 1 and the supplementary figures and the description of the measuring process support the reliability of the improved benthic Mg/Ca paleothermometer. Thus, the dataset presented in this study is reliable that is fundamental to this innovative research.

The calculation of the $\delta^{18}\text{O}$ of seawater that is thought to be a proxy of global ice volume change follows the commonly accepted method, which could stem from the work made by Nick Shackleton in 1974 and further from Cesare Emiliani and the Nobel Prize winner Harold Urey. The calculation of the sea level change using the $\delta^{18}\text{O}$ of sea water is based on Fairbanks and Matthews (1978), also a landmark in paleoceanography. Given reliable dataset and reasonable calculating method, the new results of the sea level change during M2 is robust and significant. I have no further comment on this.

If sea level really dropped as many as 55 meters during M2, marked changes in the continental shelves would be expected, such as vegetation, shift of carbonate deposition between shelf and deep sea. Could more evidences be involved in this study?

Is it possible that the BWT changes recorded in this study are local not global signals? More similar records to this study, diversely distributed in global oceans, will make the main conclusion of this study more credible.

Reviewer #2

(Remarks to the Author)

This study presents new benthic Mg/Ca-based bottom water temperature reconstructions for the M2 glaciation at sites ODP 982 and 1241, as well as for an older time period (CN4-CN6) at ODP 982. By combining these with benthic $\delta^{18}\text{O}$ data, the study also calculates $\delta^{18}\text{O}_{\text{sw}}$, providing insights into sea level fluctuations. The new reconstructions suggest a significant sea level drop of up to ~55 meters (very large!) during the M2 glaciation, which contrasts with previous findings from sites ODP 849 and U1308. T and $\delta^{18}\text{O}_{\text{sw}}$ are crucial parameters for understanding past climate change, but reliable reconstructions are essential. Here are some points for the authors to consider.

Major Points:

1. Potential carbonate ion effect on Melonis Mg/Ca. The reliance on the comparison of Mg/Ca in Melonis shells of different sizes and the carbonate chemistry profiles in pore waters (Fig. 2) is problematic. It assumes that: (1) Large and small shells inhabit different depths in pore waters (larger shells live deeper), (2) Pore-water carbonate saturation can be reliably inferred using Melonis B/Ca, and (3) Fig. 2 accurately reflects pore-water carbonate chemistry. Unfortunately, there is a lack of firm evidence supporting these assumptions. The profiles in Fig. 2 are overly simplistic and require validation with real measurements from multiple sites under various deep-water conditions (e.g., saturated vs undersaturated, high vs low organic fluxes).

2. Exceptionally large spatial gradients in $\delta^{18}\text{O}_{\text{sw}}$ during the Pliocene. The large spatial gradients in calculated $\delta^{18}\text{O}_{\text{sw}}$ (up to 1 per mil) during the Pliocene, as shown by ODP 982 and 1241 data, are confusing. In the modern ocean, the global deep ocean $\delta^{18}\text{O}_{\text{sw}}$ gradient is much smaller (< 0.3 per mil). If the reconstructions are correct, what caused the much larger interocean $\delta^{18}\text{O}_{\text{sw}}$ gradient during the Pliocene? The Pliocene BWT gradient was also much greater ($\sim 6\text{-}7^\circ\text{C}$) compared to the modern ocean ($\sim 4^\circ\text{C}$). These discrepancies require explanation, as they are critical to the reliability of the reconstructions and data interpretation.

3. Inconsistent climate signals during the M2. Fig. 5 shows a sharp increase in $\delta^{18}\text{O}_{\text{sw}}$ during the second phase of sea level drop, accompanied by rapid BWT increases at both ODP 982 and 1241. How could ice sheets grow while the climate was warming? This contradiction requires further explanation.

Detailed Comments:

1. Line 74: Do you have data for site 1143? If not, remove "1143".
2. Line 105-106: References are needed to justify the claim.
3. Line 110: Change "generally" to "sometimes" or "occasionally". In most cases, data are plotted along the 1:1 line, suggesting little difference between large and small shells. This contradicts the main text and Fig. 2.
4. Line 128: If possible at all, please provide some references using straightforward approaches (e.g., counting of shells) to support a deeper habitat for larger shells.
5. Line 128-130: Discussions about Cd/Ca and Sr/Ca are distracting and unconvincing. Consider removing these parts (e.g., Fig. S1).
6. Line 135-138: If large shells live deeper, why are they found in core-top sediments (top ~ 2 cm)? How were large shells in Fig. 1 obtained?
7. Line 144: The argument is weak; further data and discussions are needed.
8. Line 150: Remove "powerful". Calculate $\delta^{18}\text{O}_{\text{sw}}$ for core-tops and compare them with hydrographic $\delta^{18}\text{O}_{\text{sw}}$ measurements. Also, calculate and present the uncertainties.
9. Line 153-155: Specify the uncertainty (1 sigma or 2 sigma). The current uncertainty is substantially underestimated. Provide details on error calculations and compare calculated BWTs with hydrographic data.
10. Line 156-158: Analytically, similar precisions for various approaches (diluted vs undiluted). More shells would give smoother curves by minimizing the effect of inter-shell variability in Mg/Ca.
11. Line 162-180: Perform similar calculations for ODP 849 and U1308 as shown in Fig. S2f. The effect is likely small.
12. Line 182: Specify "this question".
13. Line 297: Clarify if you mean ϵNd data. Why is there no positive ϵNd excursion at ODP 1267, which is located at the upstream (SCW) of U1313?
14. Line 329-332: Provide modeling work or discuss literature modeling results to support the arguments.

Figures and Tables:

1. Table 1: Verify BWT at ODP 982 (6°C) using eWOCE or GLODAP datasets. Consider moving the table to Supplementary Information.
2. Fig. 1: Specify the thickness of core-top samples. If 1-2 cm thick, there should be no large shells. Add a cross plot: $\delta^{13}\text{C}$ vs B/Ca, comparing distributions for large vs small shells with expectations from Fig. 2.
3. Fig. 2: Justify vertical distributions with real measurements. Schematic representation is insufficient. Consider site-to-site variations.
4. Fig. 3e: If Melonis B/Ca reflects pore-water carbonate saturation, why are carbonate ion levels much lower in the Pacific than in the Atlantic? This has implications for core-top data interpretation.
5. Fig. 4: Show BWT and $\delta^{18}\text{O}_{\text{sw}}$ along the same y-axis. For example, consider adding a constant value (e.g., $+6^\circ\text{C}$) to BWT at ODP 1241 for better show spatial gradient comparison, and remember to show " $+6^\circ\text{C}$ " in the legend. Reference the latest literature BWT/ $\delta^{18}\text{O}_{\text{sw}}$ data from Rohling et al. (2022), due to potential issues with older reconstructions.
6. Fig. 5: Provide evidence for the robustness of age controls for detailed temporal resolution comparisons. As for Fig. 4, use the same y-axis scale for different locations, including ϵNd data (panel f). Distinguish yellow and orange bands (which are currently difficult to distinguish) with different colors.
7. Fig. S4: Use the same y-axis scale for $\delta^{18}\text{O}_{\text{sw}}$ at various sites.
8. Fig. S5: Consider moving to the main text. Also, Add a plot/panel: Mg/Ca vs deep-water DCO_3 .
9. Supplementary Information: Add references supporting that Melonis B/Ca reflects pore-water carbonate saturation state.

Version 1:

Reviewer comments:

Reviewer #1

(Remarks to the Author)

Thank you for carefully responding to my comments and making adequate revision.

Reviewer #2

(Remarks to the Author)

Re-review of "Major sea level fall during the Pliocene M2 glaciation" by Yang et al.

I thank the authors for taking effort to address my concerns/comments. The manuscript has been improved significantly.

However, I continue to have the following (relatively straightforward) points/concerns which need to be addressed before publication.

1. Provide the complete data set (including foram Mg/Ca alongside hydrographic data including BWT, deltaCO₃, and CO₃) for the core-top samples in the supplementary Excel data file, which is more accessible than a table in PDF;
2. Make a plot to show core-top Mg/Ca vs deltaCO₃ for the main text. I stress this because the author did not provide such a plot although requested during the first-round review. I understand the authors' preference, but it is still debated regarding whether benthic Mg/Ca is affected by deltaCO₃ as discussed in the main text. In any case, it is useful to be transparent by showing the core-top Mg/Ca vs deltaCO₃ figure in the main text (can be a new panel in Fig. 3);
3. Provide a map or maps/transects in SI to showing cores discussed in the main text. For example, it will be much easier to tell locations of cores in relation to various water masses (e.g., ODP 1267 and U1313). Also, it will be useful to add site location information (lat., lon., water depth) in the main text where appropriate.

Comments from Reviewer #1:

Firstly, we thank Reviewer 1 for their positive comments acknowledging the significant advance we have made in Mg/Ca palaeothermometry.

“If sea level really dropped as many as 55 meters during M2, marked changes in the continental shelves would be expected, such as vegetation, shift of carbonate deposition between shelf and deep sea. Could more evidences be involved in this study?”

Response: IRD related evidence of increased ice volume in both North and South hemisphere at MIS M2 are provided in Lines 854-858, and we also added more evidence of eolian dust deposition in the North Atlantic (Lang et al., 2014), biomarker records in Mojave Desert (People et al., 2024) and vegetation variability in the Arctic-Atlantic gateway region (Khan et al., 2022) in Lines 858-861. All of this evidence points to a cold climate, consistent with glacial conditions in the northern hemisphere.

Is it possible that the BWT changes recorded in this study are local not global signals? More similar records to this study, diversely distributed in global oceans, will make the main conclusion of this study more credible.

Response: Reviewer 1 is absolutely correct that individual records reflect a combination of local and global conditions. This is why, from the outset, we choose to use two sites bathed by different water masses (there are unfortunately very little existing high resolution bottom water temperature data for comparison). We have now mentioned this in lines 750-752 for clarity. As expected, the temperatures recorded by our shallower North Atlantic Site and deeper Equatorial Pacific Site have different absolute values reflecting their different source regions (Figure 4), however, they share very similar trends. Furthermore, the calculated oxygen isotopic compositions of seawater at each site also contain very similar timing and magnitude of changes to each other, which would be difficult to explain through local water mass effects at each site. Nevertheless, we do also compare our records with an inferred deep sea temperature record based on Mediterranean sea level estimate in Lines 790-793 and Figure 5.

The Mediterranean sea level estimate is derived through a completely independent approach (a sea level control on the surface salinity of the Mediterranean basin). This means that if all three records suggest a similar timing and magnitude of change, we can be more confident that the reconstructed sea level changes are robust. Indeed, we use the detailed similarities and differences between these records to discuss our relative confidence in the sequence of reconstructed sea level changes through the M2 event (Lines 790-853).

Reviewer #2 (Remarks to the Author):

We also thank Reviewer 2 for offering these great comments, which help a lot in improving this manuscript.

Major Points:

1. Potential carbonate ion effect on *Melonis* Mg/Ca. The reliance on the comparison of Mg/Ca in *Melonis* shells of different sizes and the carbonate chemistry profiles in pore waters (Fig. 2) is problematic. It assumes that: (1) Large and small shells inhabit different depths in pore waters (larger shells live deeper), (2) Pore-water carbonate saturation can be reliably inferred using *Melonis* B/Ca, and (3) Fig. 2 accurately reflects pore-water carbonate chemistry.

Unfortunately, there is a lack of firm evidence supporting these assumptions. The profiles in Fig. 2 are overly simplistic and require validation with real measurements from multiple sites under various deep-water conditions (e.g., saturated vs undersaturated, high vs low organic fluxes).

Response: After careful consideration of this major comment and related detailed comments below, we have decided to remove Fig. 2, which was intended as a simplified cartoon representing our interpretation of the controls on *Melonis* $\delta^{13}\text{C}$ and B/Ca, which suggests that at some sites, larger individuals of *Melonis* lived at slightly deeper depths in the sediment. We have replaced this with measured porewater profiles in a new Fig. 2, and added evidence in the “Improving the benthic Mg/Ca palaeothermometer” part of the discussion part 1 as follows:

i. Observed distributions of living *Melonis* spp. and *Uvigerina* spp. in the sediment column. [Due to the limited number of studies focusing on *Melonis* spp., we also included studies of other intermediate infaunal species such as *Uvigerina* spp. (Jannink et al., 1998; Rathburn and Corliss, 1994; Schmiedl et al., 2004; Theodor et al., 2016)]. These studies of living *Melonis* spp. and *Uvigerina* spp. found that individuals with larger test sizes have deeper average depth habitats than smaller individuals. To avoid misunderstanding, we use the phrase ‘average habitat depth’ in the manuscript. Lines 151-154.

ii. We can be quite confident that variations in *Melonis* B/Ca reflect porewater ΔCO_3^{2-} . There is a linear relationship between benthic foraminiferal (including *Melonis* spp.) B/Ca ratios and bottom water ΔCO_3^{2-} (Rae et al., 2011; Yu and Elderfield, 2007), this study (Lines 346-354, Fig. S2). Furthermore, there is a linear relationship between bottom water ΔCO_3^{2-} and pore water ΔCO_3^{2-} (Weldeab et al., 2016) (Lines 343-346; Fig. S1).

iii. We agree that the original cartoon used for Fig 2 was perhaps overly simplistic. We have therefore replaced it with measured porewater profiles of DIC $\delta^{13}\text{C}$ and ΔCO_3^{2-} (Fig. 2). The reviewer is correct that porewater profiles can be variable and complicated, and we believe that this figure is a better approach to illustrate the microhabitat effects on the geochemistry of infaunal foraminifera (Fig. 2).

2. Exceptionally large spatial gradients in $\delta^{18}\text{O}_{\text{sw}}$ during the Pliocene. The large spatial

gradients in calculated $\delta^{18}\text{O}_{\text{sw}}$ (up to 1 per mil) during the Pliocene, as shown by ODP 982 and 1241 data, are confusing. In the modern ocean, the global deep ocean $\delta^{18}\text{O}_{\text{sw}}$ gradient is much smaller (< 0.3 per mil). If the reconstructions are correct, what caused the much larger interocean $\delta^{18}\text{O}_{\text{sw}}$ gradient during the Pliocene? The Pliocene BWT gradient was also much greater ($\sim 6\text{-}7^\circ\text{C}$) compared to the modern ocean ($\sim 4^\circ\text{C}$). These discrepancies require explanation, as they are critical to the reliability of the reconstructions and data interpretation.

Response: The water depth of Site 982 is 1143 m, which means that Site 982 is not a deep ocean site (Table S1), and not suitable for global deep ocean gradient comparison. That is the reason that we did not discuss the BWT and $\delta^{18}\text{O}_{\text{sw}}$ offsets between Sites 982 and 1241.

The modern GLODAP database (Lauvset et al., 2024)) suggests a BWT for Site 982 of $\sim 6^\circ\text{C}$ and for ODP Site 1241 of $\sim 2^\circ\text{C}$ (Table S1). The modern BWT offset between ODP Site 982 and 1241 is $\sim 4^\circ\text{C}$, and the modern $\delta^{18}\text{O}_{\text{sw}}$ offset is $\sim 0.5\text{‰}$ (Site 982: $\sim 0.4\text{‰}$; Site 1241: $\sim -0.1\text{‰}$) (Breitkreuz et al., 2018). A larger offset in both BWT and $\delta^{18}\text{O}_{\text{sw}}$ would be expected in the Pliocene, since ODP Site 982 was influenced by warmer and saltier Mediterranean Outflow Water during the mid Pliocene (3.63 - 2.75 Ma) (Khélifi et al., 2014).

3. Inconsistent climate signals during the M2. Fig. 5 shows a sharp increase in $\delta^{18}\text{O}_{\text{sw}}$ during the second phase of sea level drop, accompanied by rapid BWT increases at both ODP 982 and 1241. How could ice sheets grow while the climate was warming? This contradiction requires further explanation.

Response: The relationship noted by the reviewer is caused by the lag of ice volume ($\delta^{18}\text{O}_{\text{sw}}$) relative to bottom water, sea surface and atmospheric temperature. This relationship has been reported from Holocene to Pliocene, and reflects the slow ice sheet response to forcings (Mudelsee, 2001; Shackleton, 2000; Shakun et al., 2015; Sosdian and Rosenthal, 2009). We have clarified the explanation of this in Lines 784-788.

Detailed Comments:

1. Line 72: Do you have data for site 1143? If not, remove “1143”.

Response: We originally included data from ODP Site 1143 samples in Fig. 1, but we have decided not to use trace metal data from this site as it has high ratios of Mn/Ca ($\sim 2000 \mu\text{mol/mol}$) and Fe/Ca ($\sim 1000 \mu\text{mol/mol}$) which indicate a potential contaminant coating.

2. Line 102-103: References are needed to justify the claim.

Response: The references for this claim are now provided in Lines 138-141.

3. Line 107: Change “generally” to “sometimes” or “occasionally”. In most cases, data are plotted along the 1:1 line, suggesting little difference between large and small shells. This contradicts the main text and Fig. 2

Response: We have changed our description for Fig. 1 (Lines 108-112 and also Lines 354-356).

4. Line 125: If possible at all, please provide some references using straightforward approaches (e.g., counting of shells) to support a deeper habitat for larger shells.

Response: We have added relevant references in Lines 151-154.

5. Line 125-127: Discussions about Cd/Ca and Sr/Ca are distracting and unconvincing. Consider removing these parts (e.g., Fig. S1).

Response: We agree that the discussion is clearer without these additional trace metals. We have therefore deleted the discussions about Sr and Cd in the Supplementary Information.

6. Line 133-136: If large shells live deeper, why are they found in core-top sediments (top ~2 cm)? How were large shells in Fig. 1 obtained?

Response: First, we revised the manuscript and now use ‘average habitat depth’ to avoid misunderstanding that larger shells only live in deeper. Second, we have clarified in the manuscript that the samples are “core-top samples” with sediment depth of 0-2 cm or “near core-top samples” with sediment depths of 2-4 cm (Lines 974-976).

7. Line 142: The argument is weak; further data and discussions are needed.

Response: We have improved our explanation of this point. We now refer to Fig 3b, which plots the Mg/Ca residuals from the Mg/Ca-temperature calibration with bottom water ΔCO_3^{2-} . If porewater ΔCO_3^{2-} affected *Melonis* Mg/Ca through a precipitation effect, we would expect to see a relationship in this figure. We also refer to literature describing how preferential dissolution can reduce test Mg/Ca without leading to obvious dissolution features under a binocular microscope. We have added the relative evidence in Lines 612-622.

8. Line 145: Remove “powerful”. Calculate $\delta^{18}\text{O}_{\text{sw}}$ for core-tops and compare them with hydrographic $\delta^{18}\text{O}_{\text{sw}}$ measurements. Also, calculate and present the uncertainties.

Response: We have removed the word “Powerful” (Line 624).

Calculating modern $\delta^{18}\text{O}_{\text{sw}}$ is a very useful suggestion. The observational benthic

$\delta^{18}\text{O}_{\text{SW}}$ data are sparse and we found no available data within a 1° radius of studied core-top stations, so we cannot compare the measured and calculated values station by station. Instead, we chose three stations in the Norwegian Sea (corresponding to KN177-2 MC 6, 11, 14), which have identical salinity values of 34.91 psu. We used a gridded data set of annual mean $\delta^{18}\text{O}_{\text{SW}}$ estimated based on the regional $\delta^{18}\text{O}_{\text{SW}}$ to salinity relationship, which produces an estimated $\delta^{18}\text{O}_{\text{SW}}$ value of $\sim 0.32\text{‰}$ (LeGrande and Schmidt, 2006). This value is comparable with bottom $\delta^{18}\text{O}_{\text{SW}}$ data from Arctic Ocean *Polar Star* cruises stations which have a similar range of salinity (~ 34.93 psu; (Marchitto et al., 2014). We compared the averaged values generated by *Melonis* and estimated $\delta^{18}\text{O}_{\text{SW}}$ value in Lines 634-636.

9. Line 149-152: Specify the uncertainty (1 sigma or 2 sigma). The current uncertainty is substantially underestimated. Provide details on error calculations and compare calculated BWTs with hydrographic data.

Response: We have specified all the uncertainties where relevant in the manuscript (1 s.d.). Our core-top data are used to construct the Mg/Ca-BWT calibration, so it would not be meaningful to compare calculated temperatures with measured BWT at these sites. We have carefully detailed how we calculate the uncertainties (Methods section Lines 1050-1074), and have ground-truthed their value on our downcore records (Lines 1075-1080).

10. Line 152-154: Analytically, similar precisions for various approaches (diluted vs undiluted). More shells would give smoother curves by minimizing the effect of inter-shell variability in Mg/Ca.

Response: We agree that using more tests is also one of the reasons behind our lower uncertainties, and we have emphasised this in Lines 629-630. The reason we wish to retain our comment on our analytical approach is that a recent paper (Lu et al. 2024) suggests that many labs have not properly accounted for matrix effects in small foraminiferal samples thereby reducing precision of records, whereas our analytical methodology does account for these matrix effects.

11. Line 158-176: Perform similar calculations for ODP 849 and U1308 as shown in Fig. S2f. The effect is likely small.

Response: We added the calculations for Site 849 (considering the resolution in U1308, we did not show the figure at this site) in the Supplementary Information. We applied the same Mn correction on ODP Site 849 Mg/Ca, which reduced Mg/Ca by ~ 0.2 - 0.3 mmol/mol, and reduced the offsets between pre/post M2 and M2 maximum by ~ 0.1 mmol/mol. (Fig. S3; Lines 118-119 in the Supplementary Information).

12. Line 178: Specify “this question”.

Response: Changed to “To reconstruct the magnitude and timing of cooling and ice growth at M2 glaciation, we analysed *Melonis* spp. $\delta^{18}\text{O}$, Mg/Ca, $\delta^{13}\text{C}$, and B/Ca from two mid-Pliocene intervals (3.313-3.184 Ma, 4.421-4.337 Ma; Fig. 4).” (Lines 746-748).

13. Line 290-291: Clarify if you mean eNd data. Why is there no positive eNd excursion at ODP 1267, which is located at the upstream (SCW) of U1313?

Response: It's Nd isotope composition of fish debris; we added the definition in Line 893. U1313 is located upstream of the deep water mass at ODP Site 1267.

14. Line 326-328: Provide modeling work or discuss literature modeling results to support the arguments.

Response: We have now cited (Fyke et al., 2018) to support this argument (Line 801).

Figures and Tables:

1. Table 1: Verify BWT at ODP 982 (6°C) using eWOCE or GLODAP datasets. Consider moving the table to Supplementary Information.

Response: We have added the reference data set at Line 978. We have moved Table 1 to the Supplementary Information as Table S1.

2. Fig. 1: Specify the thickness of core-top samples. If 1-2 cm thick, there should be no large shells. Add a cross plot: $\delta^{13}\text{C}$ vs B/Ca, comparing distributions for large vs small shells with expectations from Fig. 2.

Response: We specified the of core-top depth intervals in Lines 974-976. The explanation of the existence of large shells is found in the response to **Major comment #1** and **Detailed comment #6**. Unfortunately, we could not pick sufficient *Melonis* specimens for both stable isotope and trace metal analysis from two size fractions for most of the samples, so the $\delta^{13}\text{C}$ and B/Ca data are not matched. However, thanks to the **major comment #1** and **detailed comments #3-7**, we revised our manuscript to be more convincing. We also note that some of these core top foraminifera were probably not living at the time of collection, and hence will have been affected by bioturbation. In other words, we could easily expect a difference of 1-2cm in either direction between the depth inhabited when alive, and the depth at which they were collected.

3. Fig. 2: Justify vertical distributions with real measurements. Schematic representation is insufficient. Consider site-to-site variations.

Response: We have revised Fig. 2 with real measurements, and added discussions about site-to-site variations in Lines 141-144, 160-164 and 361-363.

4. Fig. 3e: If *Melonis* B/Ca reflects pore-water carbonate saturation, why are carbonate ion levels much lower in the Pacific than in the Atlantic? This has implications for core-top data interpretation.

Response: According to the bottom water and porewater carbonate saturation data set of the sites across Atlantic Ocean (63 sites in total) with different latitudes and water depths, the ΔCO_3^{2-} of pore water linearly co-varies with the ΔCO_3^{2-} of overlying bottom water (Weldeab et al., 2016) (Fig. S1). Using pH and Alk values from GLODAP datasets, we calculated the modern bottom water ΔCO_3^{2-} at the adjacent stations to ODP Sites 982 and 1241 using seacarb package (Gattuso et al., 2021), the values are 55.7 and 5.5 at Sites 982 and 1241 respectively. As noted by the reviewer, Pliocene *Melonis* B/Ca ratios at ODP Site 1241 are lower than those at ODP Site 982. This indicates that just as in the modern, porewater ΔCO_3^{2-} of ODP Site 1241 was lower than at ODP Site 982. We have added the relevant discussion in Supplementary Information (Lines 142-146 in the Supplementary Information).

5. Fig. 4: Show BWT and $\delta^{18}\text{O}_{\text{sw}}$ along the same y-axis. For example, consider adding a constant value (e.g., +6oC) to BWT at ODP 1241 for better show spatial gradient comparison, and remember to show “+6oC” in the legend. Reference the latest literature BWT/ $\delta^{18}\text{O}_{\text{sw}}$ data from Rohling et al. (2022), due to potential issues with older reconstructions.

Response: After careful consideration, we prefer to keep the Fig. 4 records on their own axes (now Fig. 5 in revised manuscript). The disadvantage is visualisation of the spatial gradient, but the advantage is that the reader can understand the absolute values more easily. However, we applied this reviewer’s suggestion when revising the figures of ‘age control’ (Fig. S7) and ‘Mn-coating correction’ (Fig. S3f), as we found that using the same y-axis works well when the actual values are not important and/or the offsets between records from different sites are important.

We have now added BWT data from (Rohling et al., 2022) (Fig. 5a), but we still focus on (Rohling et al., 2014) in the discussion since the records in that study do not use the benthic foraminiferal $\delta^{18}\text{O}$ stack, so it provides a more independent support of our sea level reconstruction.

6. Fig. 5: Provide evidence for the robustness of age controls for detailed temporal resolution comparisons. As for Fig. 4, use the same y-axis scale for different locations, including eNd data (panel f). Distinguish yellow and orange bands (which are currently difficult to distinguish) with different colors.

Response: We have added a figure with benthic foraminiferal $\delta^{18}\text{O}$ records from every site used in Fig. 5 and Fig. 6 in the Supplementary Information (Fig. S7) and included a relevant explanation in the manuscript Lines 869-872.

We have changed the colour bands as suggested (Fig. 6).

For reasons explained in the response to **Figures and tables comment #5**, we prefer to keep them on separate axes.

7. Fig. S4: Use the same y-axis scale for $\delta^{18}\text{O}_{\text{sw}}$ at various sites.

Response: For reasons explained in the response to **Figures and tables comment #5**, we prefer to keep them on separate axes (Fig. S6).

8. Fig. S5: Consider moving to the main text. Also, Add a plot/panel: Mg/Ca vs deep-water DCO_3 .

Response: We moved the figure to the main text (Fig. 3a).

We have added the panel of the Mg/Ca standardized residual versus bottom water ΔCO_3^{2-} and referred to this to improve our discussion (Lines 612-618) (Fig. 3b).

9. Supplementary Information: Add references supporting that Melonis B/Ca reflects pore-water carbonate saturation state.

Response: We have added the relevant explanation in the manuscript Lines 156-159, 343-354, and also in the Supplementary Information (Fig. S1, S2).

References

- Breitkreuz, C., Paul, A., Kurahashi-Nakamura, T., Losch, M., Schulz, M., 2018. A Dynamical Reconstruction of the Global Monthly Mean Oxygen Isotopic Composition of Seawater. *J. Geophys. Res.: Oceans* 123, 7206-7219.
- Fyke, J., Sergienko, O., Löfverström, M., Price, S., Lenaerts, J.T.M., 2018. An Overview of Interactions and Feedbacks Between Ice Sheets and the Earth System. *Rev. Geophys.* 56, 361-408.
- Gattuso, J.-P., Epitalon, J.-M., Lavigne, H., Orr, J., 2021. Seacarb: Seawater carbonate chemistry. R package version 3.3.0.
- Jannink, N.T., Zachariasse, W.J., Van der Zwaan, G.J., 1998. Living (Rose Bengal stained) benthic foraminifera from the Pakistan continental margin (northern Arabian Sea). *Deep Sea Res. Part I* 45, 1483-1513.
- Khan, S., Farooqui, A., Shukla, U.K., Grøsfjeld, K., Knies, J., Prasad, V., 2022. Late Pliocene continental climate and vegetation variability in the Arctic-Atlantic gateway region prior to the intensification of Northern Hemisphere glaciations. *Palaeogeogr. Palaeoclimatol. Palaeoecol.* 586, 110746.
- Khélifi, N., Sarnthein, M., Frank, M., Andersen, N., Garbe-Schönberg, D., 2014. Late Pliocene variations of the Mediterranean outflow. *Mar. Geol.* 357, 182-194.
- Lang, D.C., Bailey, I., Wilson, P.A., Beer, C.J., Bolton, C.T., Friedrich, O., Newsam, C., Spencer, M.R., Gutjahr, M., Foster, G.L., Cooper, M.J., Milton, J.A., 2014. The transition on North America from the warm humid Pliocene to the glaciated Quaternary traced by eolian dust deposition at a benchmark North Atlantic Ocean drill site. *Quat. Sci. Rev.* 93, 125-141.
- Lauvset, S.K., Lange, N., Tanhua, T., Bittig, H.C., Olsen, A., Kozyr, A., Álvarez, M., Azetsu-Scott, K., Brown, P.J., Carter, B.R., Cotrim da Cunha, L., Hoppema, M., Humphreys, M.P., Ishii, M., Jeansson, E., Murata, A., Müller, J.D., Pérez, F.F., Schirnack, C., Steinfeldt, R., Suzuki, T., Ulfso, A., Velo, A., Woosley, R.J., Key, R.M., 2024. The annual update GLODAPv2.2023: the global interior ocean biogeochemical data product. *Earth Syst. Sci. Data* 16, 2047-2072.
- LeGrande, A.N., Schmidt, G.A., 2006. Global gridded data set of the oxygen isotopic composition in seawater. *Geophys. Res. Lett.* 33, L12604.
- Marchitto, T., Curry, W., Lynch-Stieglitz, J., Bryan, S., Cobb, K., Lund, D., 2014. Improved oxygen isotope temperature calibrations for cosmopolitan benthic foraminifera. *Geochim. Cosmochim. Acta* 130, 1-11.
- Mudelsee, M., 2001. The phase relations among atmospheric CO₂ content, temperature and global ice volume over the past 420 ka. *Quat. Sci. Rev.* 20, 583-589.
- People, M.D., Bhattacharya, T., Tierney, J.E., Knott, J.R., Lowenstein, T.K., Feakins, S.J., 2024. Biomarker Evidence for an MIS M2 Glacial-Pluvial in the Mojave Desert Before Warming and Drying in the Late Pliocene. *Paleoceanogr. Paleoclimatol.* 39, e2023PA004687.
- Rae, J.W.B., Foster, G.L., Schmidt, D.N., Elliott, T., 2011. Boron isotopes and B/Ca in benthic foraminifera: Proxies for the deep ocean carbonate system. *Earth Planet. Sci. Lett.* 302, 403-413.
- Rathburn, A.E., Corliss, B.H., 1994. The ecology of living (stained) deep-sea benthic foraminifera from the Sulu Sea. *Paleoceanography* 9, 87-150.
- Rohling, E.J., Foster, G.L., Gernon, T.M., Grant, K.M., Heslop, D., Hibbert, F.D., Roberts, A.P., Yu, J., 2022. Comparison and synthesis of sea-level and deep-sea temperature variations over the past 40 million years. *Rev. Geophys.* 60, e2022RG000775.
- Rohling, E.J., Foster, G.L., Grant, K.M., Marino, G., Roberts, A.P., Tamisiea, M.E., Williams, F., 2014.

- Sea-level and deep-sea-temperature variability over the past 5.3 million years. *Nature* 508, 477-482.
- Schmiedl, G., Pfeilsticker, M., Hemleben, C., Mackensen, A., 2004. Environmental and biological effects on the stable isotope composition of recent deep-sea benthic foraminifera from the western Mediterranean Sea. *Mar. Micropaleontol.* 51, 129-152.
- Shackleton, N.J., 2000. The 100,000-Year Ice-Age Cycle Identified and Found to Lag Temperature, Carbon Dioxide, and Orbital Eccentricity. *Science* 289, 1897-1902.
- Shakun, J.D., Lea, D.W., Lisiecki, L.E., Raymo, M.E., 2015. An 800-kyr record of global surface ocean $\delta^{18}\text{O}$ and implications for ice volume-temperature coupling. *Earth Planet. Sci. Lett.* 426, 58-68.
- Sosdian, S., Rosenthal, Y., 2009. Deep-Sea Temperature and Ice Volume Changes Across the Pliocene-Pleistocene Climate Transitions. *Science* 325, 306-310.
- Theodor, M., Schmiedl, G., Mackensen, A., 2016. Stable isotope composition of deep-sea benthic foraminifera under contrasting trophic conditions in the western Mediterranean Sea. *Mar. Micropaleontol.* 124, 16-28.
- Weldeab, S., Arce, A., Kasten, S., 2016. $\text{Mg}/\text{Ca}-\Delta\text{CO}_{3\text{porewater}}^{2-}$ – temperature calibration for *Globobulimina* spp.: A sensitive paleothermometer for deep-sea temperature reconstruction. *Earth Planet. Sci. Lett.* 438, 95-102.
- Yu, J., Elderfield, H., 2007. Benthic foraminiferal B/Ca ratios reflect deep water carbonate saturation state. *Earth Planet. Sci. Lett.* 258, 73-86.

Comments from Reviewer #1:

Thank you for carefully responding to my comments and making adequate revision.

Response: We thank Reviewer #1 for the positive feedback.

Comments from Reviewer #2:

Re-review of “Major sea level fall during the Pliocene M2 glaciation” by Yang et al. I thank the authors for taking effort to address my concerns/comments. The manuscript has been improved significantly.

Response: We thank Reviewer #2 for recognizing our efforts in addressing their valuable comments.

However, I continue to have the following (relatively straightforward) points/concerns which need to be addressed before publication.

1. Provide the complete data set (including foram Mg/Ca alongside hydrographic data including BWT, ΔCO_3 , and CO_3) for the core-top samples in the supplementary Excel data file, which is more accessible than a table in PDF;

Response: We provided the data set of core-top samples (used to be Table S3 in Supplementary Information) in the excel file ‘source data’, in the sheet named ‘Figure 3’.

2. Make a plot to show core-top Mg/Ca vs ΔCO_3 for the main text. I stress this because the author did not provide such a plot although requested during the first-round review. I understand the authors’ preference, but it is still debated regarding whether benthic Mg/Ca is affected by ΔCO_3 as discussed in the main text. In any case, it is useful to be transparent by showing the core-top Mg/Ca vs ΔCO_3 figure in the main text (can be a new panel in Fig. 3);

Response: We added a panel in Fig. 3 which is the cross plot of core-top Mg/Ca versus bottom water ΔCO_3^{2-} (Fig. 3b).

3. Provide a map or maps/transects in SI to showing cores discussed in the main text. For example, it will be much easier to tell locations of cores in relation to various water masses (e.g., ODP 1267 and U1313). Also, it will be useful to add site location information (lat., lon., water depth) in the main text where appropriate.

Response: We added a figure in Supplementary Information which includes the map of studied and discussed sites in the main text, and also a transect in Atlantic to show the

vertical distribution of water masses and the position of those sites (Fig. S3). We also added location information for the discussed sites in the main text (Line 208-209, 328-330).